

# Numerical study of the initial condition and emission on simulating PM₂.₅ concentrations in Comprehensive Air Quality Model with extensions version 6.1 (CAMx v6.1): Taking Xi'an as example

Han Xiao [1], Qizhong Wu [1*], Xiaochun Yang [1,2], Lanning Wang[1], Huaqiong Cheng[1]

[1] College of Global Change and Earth System Science, Beijing Normal University, Beijing 100875, China
[2] Xi'an Meteorological Bureau, Xi'an, Shaanxi Province 710016, China

*Correspondence to*: Qizhong Wu (wqizhong@bnu.edu.cn)

**Abstract.** A series of model sensitivity experiments is designed to explore the effects of different initial conditions and emissions in Xi'an in December 2016, which is a major city in the key area "Fen-Wei Plains" for air pollution control in China. Three methods were applied for the initial condition tests: clean initial mechanism, restart mechanism and continuous simulation. In clean initial mechanism test, the sensitivity experiments C00, C06, C12, C18, and C24 were conducted according to the intercepted time periods, and the results showed that the model performance of PM₂.₅ was better with the delay of the start time of the intercepted time periods. From experiments C00 to C24, the absolute mean bias (MB) decreased from 51.07 μg/m³ to 3.72 μg/m³ and the index of agreement (IOA) increased from 0.49 to 0.86, which illustrates that the model performance of C24 is much better than C00. In order to explore the restart mechanism, sensitivity experiments R1120 and R1124 were set according to the time of the first day for the model simulation. Although the start times of simulations were different, after a period of spin-up time, the simulation results with different start time were nearly consistent, the results showed that the spin-up time is about 27 hours. As for the continuous simulation test, CT12 and CT24 were conducted. The start time of the intercepted time periods for CT12 and R1120 were the same, and the simulation results were nearly identical. The simulation results of CT24 performed best in all the sensitivity experiments, with the correlation coefficient (R), MB, and IOA reaching 0.81, 6.29 μg/m³, and 0.90, respectively. For the emission tests, the updated local emission inventory with construction fugitive dust emissions have been added and compared to the simulation results of the original emission inventory. The simulation with the updated local emissions showed a much better performance on PM₂.₅ modelling. Therefore, combining the method of CT24 with the updated local emission inventory can nicely improve the model performance of PM₂.₅ in Xi'an, the absolute MB decreased from 35.16 μg/m³ to 6.29 μg/m³ and the IOA reached 0.90.

## 1 Introduction

In the recent years, severe air pollution has gradually become a big challenge in China and other developing countries (Wu et al., 2014; Li et al., 2017). China released three-year action plan for cleaner air in 2018, and efforts will be focused on areas including the Beijing-Tianjin-Hebei region, the Yangtze River Delta, and the Fen-Wei Plains. As a major city in the key area "Fen-Wei Plains", Xi'an is located in the Guanzhong Basin. The city is surrounded by the Qinling Mountains to the south, and the Loess Plateau extends to the north and west, which is not conductive to the spread of air pollutant. Xi'an has suffered





serious air pollution in recent years because of its special topography and rapid economic development (Zhang, et al., 2002; Cao, et al., 2012). Worse, Xi'an is under rapid development of urban construction activities with large construction fugitive dust (Long et al., 2016).

Air quality modelling systems is an important tool for air pollution assessment and have evolved over three generations since the 1970s, driven by crucial regulations, societal and economic needs, and increasing high performance computing capacity (Zhang et al., 2012). Various air quality models are widely used in the simulation and forecasting of pollutants, such as the Community Multiscale Air Quality (CMAQ) (Eder and Yu, 2006; Appel et al., 2017), the Comprehensive Air Quality Model with extensions (CAMx) (ENVIRON, 2013), WRF-Chem (Grell et al., 2005), and the Nested Air Quality Prediction

Modeling System (GNAQPMS/NAQPMS) (Wang et al., 2006; Chen et al., 2015; Wang et al., 2017). In order to accurately analyze the apportionment of emission categories and contributions from different source regions for atmospheric pollutions, many researchers used the CAMx model with the particulate matter source apportionment technology (PSAT) in different areas of China, including Beijing (Zhang et al., 2018), Tangshan (Li et al., 2013), Pearl River Delta region (Wu et al., 2013) and Yangtze River Delta region (Li et al., 2011). The CAMx showed good model performances for simulation of air pollution

(Panagiotopoulou et al., 2016).

    The input files for CAMx model include initial/boundary conditions, gridded and elevated point source emissions, and meteorological files (ENVIRON, 2013). The meteorology and emissions inputs can cause great uncertainty for air quality models (Tang et al. 2010; Gilliam et al., 2015). Many researchers reduced the uncertainty of meteorology through refined physical parameterizations and other techniques such as data assimilation (Sistla et al., 1996; Seaman, 2000; Gilliam et al.,

2015; Li et al., 2019). A reasonable emission inventory is very important for the simulation accuracy of the air quality model. Many researchers studied East Asian emissions (Kato et al., 1992; Streets et al., 2003; Ohara et al. 2007; Zhang et al., 2009;), and tried to construct emission inventories of particulate matter (PM) in China (Wang et al. 2005; Zhang et al. 2006). However, the absence of detailed information on China introduces great uncertainty into the emission inventories (Cao et al., 2011). In recent years, more and more researchers focused on constructing and updating of regional local emission inventories to

improve the model performance. Wu et al. (2014) improved the model performance by adding more regional point source emissions and updating the area source emissions in village and surrounding cities of Beijing. Based on that work, Yang et al. (2019) have added local datasets into the emission inventory of Guanzhong Plain, China, which was applied in simulating $PM_{2.5}$ concentrations by CMAQ model in Xi'an. Numerous works indicated that construction dust emission plays an important role of the air pollution, especially in urban areas (Ni et al., 2012; Huang et al., 2014; Wang et al., 2015). In our previous study,

we built a particulate matter emission inventory from construction activities at county level in Xi'an, based on an extensive survey of construction activities and combining with two sets of dust emission factors for the typical city in north China (Xiao et al., 2019).

    However, few studies have investigated the effects of initial condition on the simulation or prediction of $PM_{2.5}$ concentrations. Therefore, the purpose of this study was to explore the effects of different initial mechanism and emissions on

the model performance of $PM_{2.5}$ in the CAMx model. A series of model sensitivity experiments for the initial mechanism and





emissions are designed to find a suitable method for simulating PM$_{2.5}$ concentrations with the reasonable initial condition and emission inventory. Not only for Xi'an, other cities may apply the similar research method using for simulating PM$_{2.5}$ concentrations in the future.

The paper is organized as follows. Section 2 gives the model descriptions for WRF-SMOKE-CAMx model system, including meteorological fields, air quality model descriptions, model domain, emission inventory and processes in Sects 2.1-2.4. Section 3 presents the sensitivity experiments design of different initial condition and emission. Section 4 discuss the model performance of the initial condition tests and emission tests to simulate the PM$_{2.5}$ concentration model in Xi'an. The conclusions are given in Section 5.

## 2 WRF-SMOKE-CAMx model descriptions

In this study, the National Center for Atmospheric Research (NCAR) Weather Research and Forecasting (WRF v3.9.1.1) model (Skamarock et al., 2008), the Center for Environmental Modeling for Policy Development (CEMPD) Sparse Matrix Operator Kernel Emissions (SMOKE v2.4) (Houyoux and Vukovich, 1999) and the Ramboll Environ Comprehensive Air Quality Model with Extensions (CAMx v6.1) (ENVIRON, 2013) were used to build up the air quality model system as shown in Fig.1. The WRF model provided the meteorological conditions for the SMOKE and CAMx model. And the SMOKE model was used to process the emissions data and provide 4-D, model-ready gridded emissions for the air quality model CAMx.

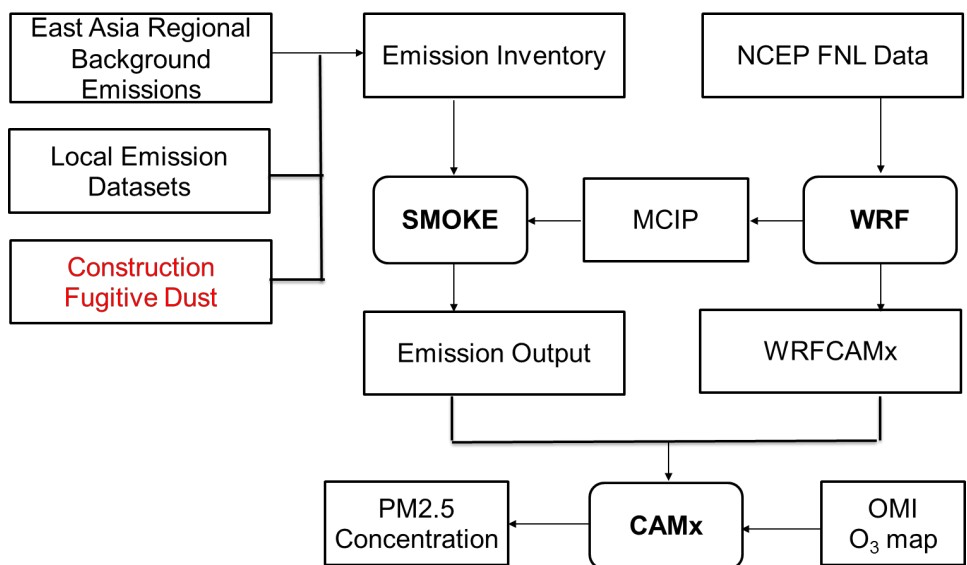

**Figure 1.** The framework of the WRF-SMOKE-CAMx model system in Xi'an. OMI O$_3$ map prepares ozone column input files for CAMx to improve the photolysis rate calculation. The CAMx forecasted the air pollutant for the next 48 hours.



## 2.1 Meteorological fields

For the WRF model configuration, we chose the rapid radiative transfer model (RRTM; Mlawer et al., 1997) and the Dudhia for longwave and shortwave radiation options (Dudhia, 1989), WSM3 cloud microphysics (Hong et al., 2004), the YSU scheme (Hong et al., 2006), the Kain–Fritsch (new Eta) cloud parameterization (Kain, 2004), and 5-layer thermal diffusion scheme (Dudhia, 1996). The meteorological initial and boundary conditions were derived from the National Centers for Environmental Prediction (NCEP) global final analysis data (FNL), with spatial resolution of $1° \times 1°$ and temporal resolution of 6-h. The

simulation was performed for the period of November 20, 2016 to January 20, 2017.

The simulation effect of daily average temperature (T2) and relatively humidity (RH2) simulated by WRF model in domain 3 were primarily validated by the observation data at 7 monitoring stations in Xi'an, which the station map was shown in Fig.2. Some statistical parameters of Appendix A were used to evaluate the model performance and shown in Table 1, and the time series was shown in Fig 3. The ME, R and RMSE of daily average T2 are 1.37℃, 0.80, 1.65℃, respectively, and the simulation

shows cooling bias of -0.95℃. The ME and RMSE of daily average RH2 are 6.77% and 8.30%. The correlation coefficient of the relatively humidity is 0.71, which is reasonable. RH2 was slightly overestimated as the MB is 6.22%.

In previous studies, Yang et al. (2019) used WRF to drive CMAQ model for winter air quality in Xi'an, and the model evaluations for winter in 2016 showed that the MB, ME, R and RMSE of T2 were -2.83℃, 2.83℃, 0.89, 3.29℃, respectively. And the MB, ME, R and RMSE of RH2 were 9.59%, 10.63%, 0.71, 13.43%, respectively. Wu et al. (2010) used the fifth-

generation NCAR/Penn State Mesoscale Model (MM5) as meteorological driver for the Nested Air Quality Prediction Modeling System (NAQPMS), and the statistical results showed that the MB and R of T2 were 2.1℃ and 0.84 and those of RH2 were -15.8% and 0.65.

Compared with previous studies, T2 and RH2 in this study have lower MB, ME, and RMSE. The R of T2 is slightly lower than previous studies, while the R of RH2 is higher. Thus, the meteorological simulation of this study is reasonable.

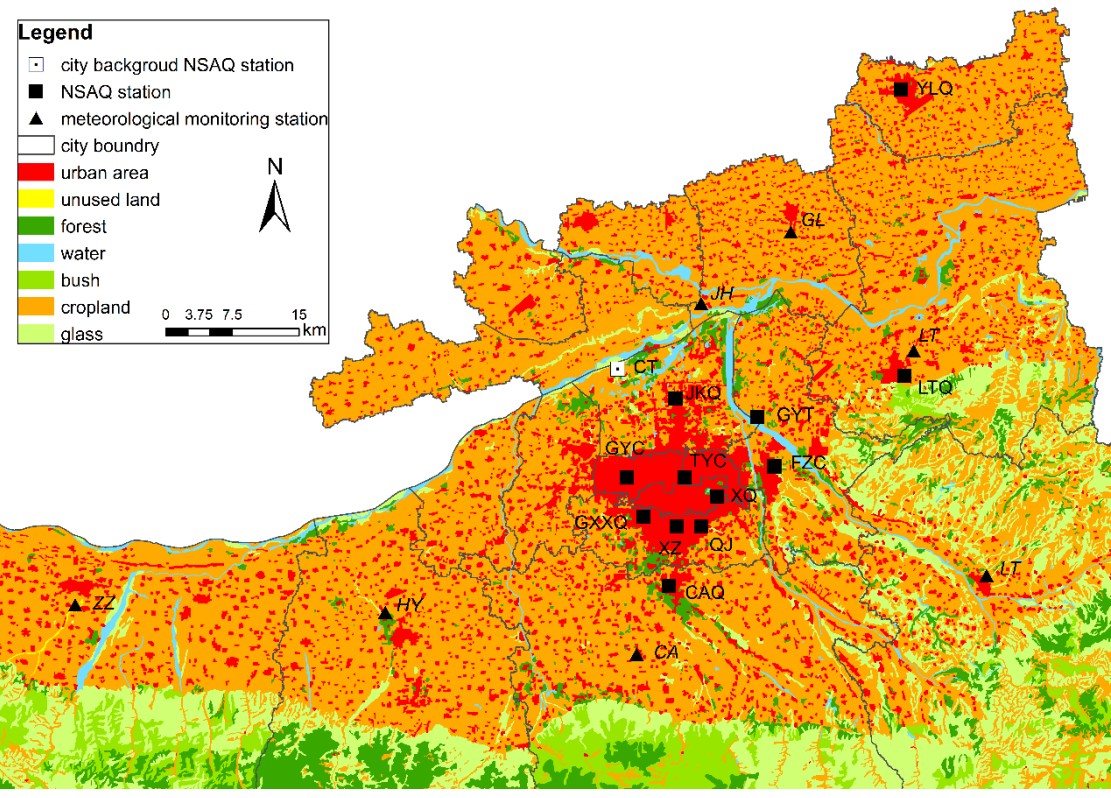


**Figure 2.** The stations map of the meteorological and air quality monitoring network in Xi'an. The triangles are the meteorological monitoring stations. The square with dot is the city background station and the black squares are the National Standard Air Quality Observation (NSAQ) Stations: Gaoyachang (GYC), Xingqing (XQ), Fangzhicheng (FZC), Xiaozhai (XZ), Tiyuchang (TYC), Gaoxinxiqu (GXXQ), Jingkaiqu (JKQ), Qujiang (QJ), Gaoyuntan (GYT), Changanqu (CAQ),

Yanliangqu (YLQ), Lintongqu (LTQ), and Caotan (CT) Station.

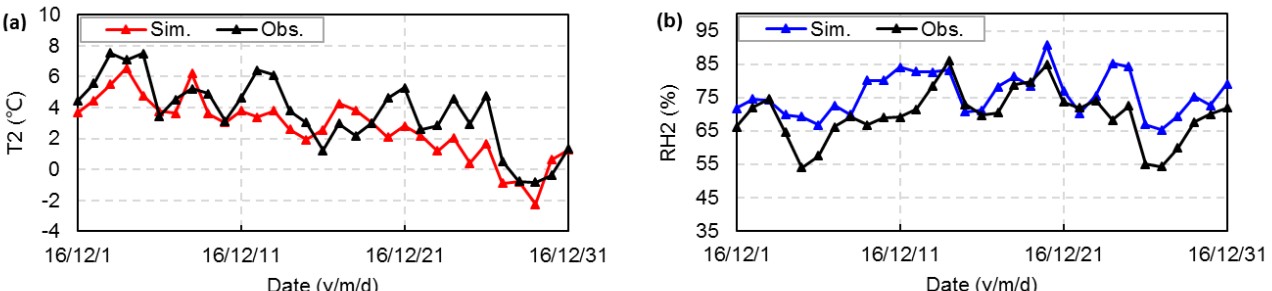

**Figure 3.** Time series plots of **(a)** daily average simulated and in situ 2m temperature (T2) as well as **(b)** simulated and in situ 2m relatively humidity (RH2) at the Xi'an station.



**Table 1.** Verification statistics of daily temperature at 2m height (T2), relatively humidity at 2m height (RH2).

| Variable | Mean | | ME | MB | R | RMSE |
|---|---|---|---|---|---|---|
| | Obs. | Sim. | | | | |
| T2(℃) | 3.68 | 2.73 | 1.37 | - 0.95 | 0.80 | 1.65 |
| RH2(%) | 69.65 | 75.88 | 6.77 | 6.22 | 0.71 | 8.30 |

## 2.2 Air quality model descriptions

The CAMx model is a state-of-the-science air quality model, which is developed by the Ramboll Environ
(http://www.camx.com). In this study, the PPM advection scheme (Colella and Woodward, 1984) is used for horizontal diffusion, and the K-theory is selected for vertical diffusion. The Regional Acid Deposition Model (RADM-AQ) (Chang et al., 1987) scheme as the aqueous-phase oxidation, ISORROPIA (Nenes et al., 1999) as inorganic aerosol thermodynamic equilibrium, and CB05 (Yarwood et al., 2005) as the gas-phase chemical mechanism, and the Euler-Backward Iterative (EBI) solver with Hertel's solutions (Hertel et al., 1993) is used in the model system. The resistance model for gases (Zhang et al., 2003) and aerosols (Zhang et al., 2001) in dry deposition module, and scavenging model for gases and aerosols (Seinfeld and Pandis, 1998) in wet deposition module is chosen in this study. The CAMx model forecasted the next 48 hours' $PM_{2.5}$ concentrations in clean initial mechanism testing and will be described more in Sec. 2.3. On the first day, CAMx used the results of the ICBCPREP which can prepare simple, static CAMx initial condition (IC) and boundary condition (BC). On the following days, it used the different initial conditions of the sensitivity experiments.

## 2.3 Model domain

Three-nest domains were designed for the WRF model (Fig. 4), with a horizontal resolution of 27 km × 27 km (D1), 9 km × 9 km (D2) 3 km × 3 km (D3), respectively. The biggest domain (D1) covered the most parts of China, the second domain (D2) includes Shaanxi Province, Shanxi Province, Henan Province and the inner domain (D3) focused on the 11 cities in Fen-wei Plain, including Xi'an. The CAMx has only one domain and the settings are the same as those in the D3 domain, while focusing on Xi'an as one sensitivity test area for initial conditions and emissions. To reduce the boundary effects, the CAMx model cut down the outermost grid of WRF model and used the variable of the center grid in WRFCAMx module, thus, the CAMx model was three grid cells smaller than the WRF model in D3 domain. The vertical resolution of WRF was 37 layers from the ground to 5 hPa at the top, and 14 layers were extracted by the WRFCAMx module, which can convert the WRF output files into the data format for CAMx model.



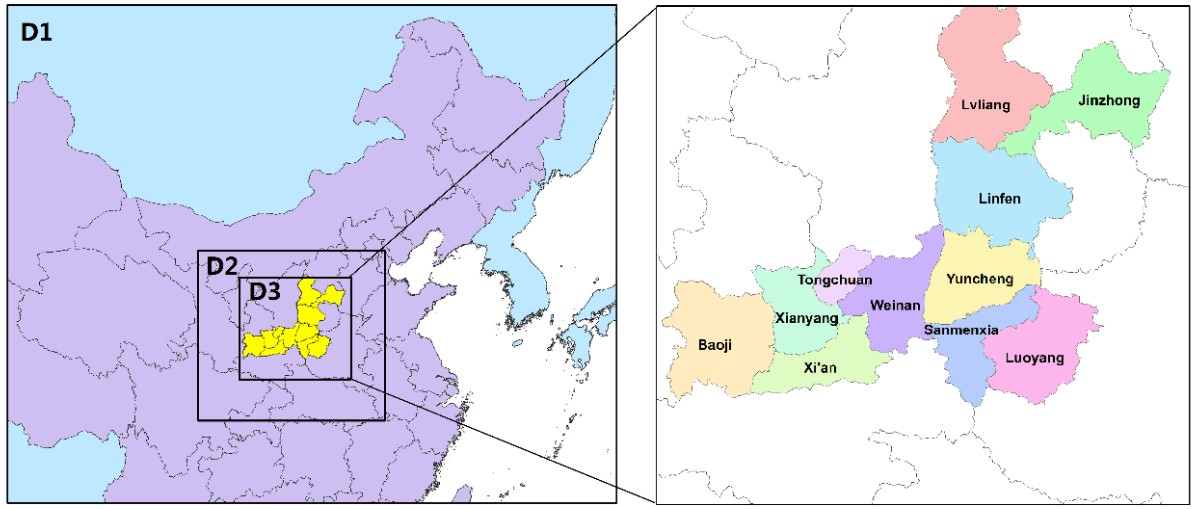

**Figure 4.** The three-nest model domain with 27-9-3 km horizontal resolution in the WRF-CAMx modelling system. D1 covers most parts of China, with 148 × 121 grids, and D2 includes Shaanxi, Shanxi and Henan Provinces. The inner domain covers Fen-wei Plain, including Xi'an.

### 2.4 Emission Inventory and Processes

The SMOKE version 2.4 (Houyoux and Vukovich, 1999) model was used to improve the Fen-wei emissions, especially Xi'an local emissions, and provide the gridded emissions for CAMx model in this study. Based on the emission inventories of previous study (Yang et al., 2019), this study added the emissions quantity of $PM_{2.5}$ from construction fugitive dust in Xi'an to update the local emission inventories. The emission inventories in this study including:

1. The regional emissions in East Asia and the local emissions in Guanzhong plain were obtained from Wu et al. (2014) and Yang et al. (2019). Major industrial emissions had a little adjusted according the annual report in this study. The emission inventory in city-level is presented in Table.2.

2. Construction fugitive dust emissions in Xi'an, based on the survey data of construction projects in Fig. 5, were collected in the previous study (Xiao et al.,2019), indicated as "local area source". This is new dataset at the county level and update in the 2017. The basic data included the location and area of each construction project. Also we replenished the missing construction data and correct the error information with Google Earth and other geographic information tools to get more accurate location information. According to statistics, there were 1595 construction projects in Xi'an in 2017, with 86.1km² of the total construction area. The construction area in the main urban (Xincheng, Beilin, Lianhu, Yanqiao, Weiyang and Yanta) was about 62.2km², accounting for 7.5% of the total area in the main urban. The distribution of the construction fugitive dust emissions in Xi'an is shown in Fig. 6.





We took the statistics-allocation approach to generate gridded area source emissions, which was to allocate the total emissions to each horizontal model grid according to the related spatial factors. In this study, the Land Scan 2015 Global Population Database (Dobson et al., 2000) was used as a spatial factor of population to allocate the emissions. For the construction fugitive dust emissions, we used the area of each construction project as the weight in the surrogate calculation,

allocated the input construction projects data to the target polygons (map of administrative division in Xi'an at the county level) based on weighted spatial overlap of the input data and target polygons. And the spatial results provide to the SMOKE model as spatially allocated factor. The horizontal and vertical allocation of point source emissions were assigned from their longitude-latitude coordinates and the Briggs algorithm (Briggs, 1972; 1984), respectively. While the temporal variation and chemical species allocation were based on profile files in SMOKE model.

As shown in Table 2, the NOx emissions ranged from 352.0 kt yr$^{-1}$ to 758.5 kt yr$^{-1}$ between 2008 in Zhang et al. (2009) to 2017 in this study. For $PM_{10}$ emissions in Shaanxi Province, the emissions also increased from 474.0 kt yr$^{-1}$ to 830.0 kt yr$^{-1}$. $PM_{10}$ emission in this study are higher than others, because including the construction fugitive dust. Other emission species such as NOx, $SO_2$, $NH_3$, VOCs and CO are little higher in this study than previous studies.

**Table 2.** Emission of major anthropogenic species in Shaanxi Province (Unit: $10^3$ tons yr$^{-1}$).

|  |  | CO | NO$_X$ | VOCs | NH$_3$ | SO$_2$ | PM$_{10}$ | PM$_{2.5}$ |
|---|---|---|---|---|---|---|---|---|
|  | point source | 1196.0 | 534.4 | 1572.7 | - | 724.7 | 321.7 | 257.5 |
|  | area source | 3272.5 | 224.1 | 471.9 | 294.0 | 490.2 | 508.3 | 244.9 |
|  | Xi'an | 964.1 | 177.5 | 370.5 | 23.4 | 155.4 | 198.6 | 82.8 |
| This study | Baoji | 628.3 | 65.8 | 256.9 | 32.8 | 131.0 | 68.4 | 41.1 |
|  | Xianyang | 773.9 | 93.2 | 584.5 | 25.9 | 173.0 | 88.6 | 66.8 |
|  | Tongchuan | 80.6 | 45.0 | 32.2 | 4.4 | 27.5 | 60.5 | 32.3 |
|  | Weinan | 561.9 | 140.3 | 500.9 | 30.5 | 224.7 | 132.6 | 103.7 |
|  | Shaanxi Prov. | 4468.5 | 758.5 | 2044.6 | 294.0 | 1214.8 | 830.0 | 502.3 |
| Zhang et al. 2009 | Shaanxi Prov. | 3528.0 | 352.0 | 491.0 | - | 907.0 | 474.0 | 328.0 |
| CCCPSC, 2011 | Shaanxi Prov. | - | 521.2 | - | - | 938.7 | 580.1 | - |
| Yang et al. 2019 | Shaanxi Prov. | 4369.0 | 736.9 | 1994.1 | 293.2 | 1193.7 | 770.4 | 534.9 |
| Yang et al. 2020 | Shaanxi Prov. | 3905.8 | 575.7 | 1904.3 | 287.6 | 802.3 | 564.0 | 398.1 |

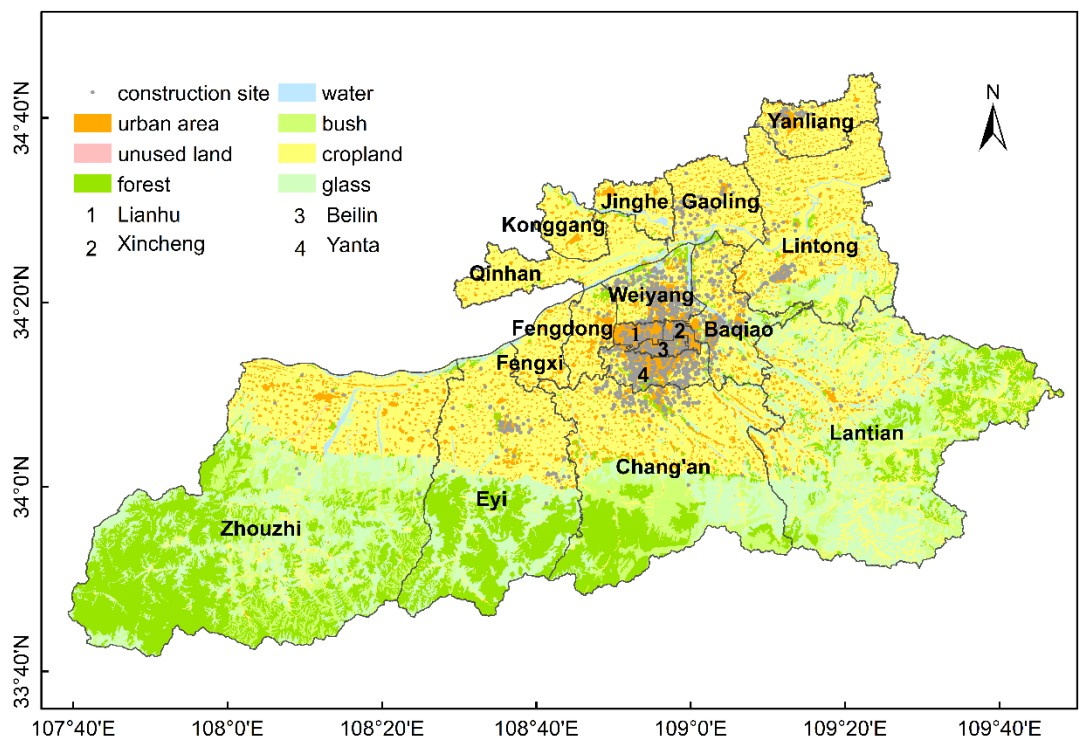

**Figure 5.** Spatial distribution of construction sites in Xi'an. Gray dots indicate the construction sites. The base map shows the types of land use (Xiao et al., 2019).


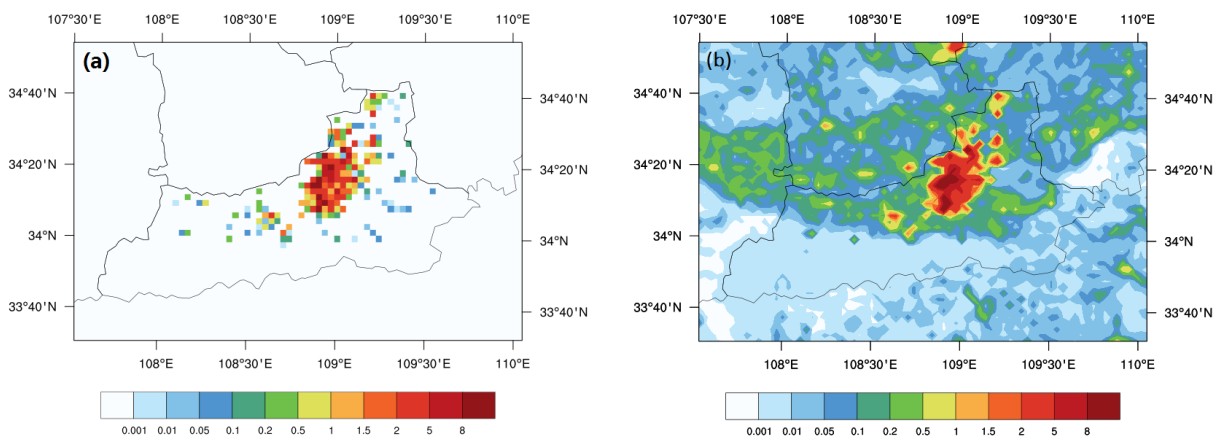

**Figure 6.** The spatial distribution of PM$_{10}$ emissions in Xi'an and its surrounding area. (a) only construction fugitive dust in Xi'an. (b) all surface PM$_{10}$ emissions in Xi'an. The grid size is 3 km x 3 km. Unit: g/km²·s





## 3 Sensitivity experiments design

A set of model sensitivity experiments under different initial conditions and emissions are designed in this study. Three
methods were applied for the initial condition tests: using the clean initial condition files as clean initial mechanism, using the
restart files as restart mechanism and the continuous simulation. For the emission tests, we compared the simulation results of
the original emission inventory and the updated local emission inventory with construction fugitive dust emissions. The
configurations of the simulation sensitivity experiments are shown in Table 3, and the time period for each initial condition
experiments are shown in Fig. 7.

### 3.1 ICON test for using the clean initial condition files

For using the clean initial condition files, icbcprep module used a clean-troposphere vertical profile to generate the initial
concentration fields for each day of simulation. The output files of CAMx were initialized at 13:00 UTC. The CAMx model
forecasted the next 48 hours' $PM_{2.5}$ concentrations in each cycle simulation. By extracting data from simulated results based
on different time periods (0~24h, 6~30h, 12~36h, 18~42h and 24~48h, respectively) shown in Fig. 7 (a), we conducted the
sensitivity experiments C00, C06, C12, C18, and C24, and explore the influence of different time periods on the simulation
effect of $PM_{2.5}$. For the sensitivity experiment C00, the data of the period for the first 24 h of output file was cut and merged
to analyze. And for C06, the first 6 h of data was spin-up time, we cut and merged the data of the period from 19:00 UTC to
18:00 UTC in the second day. C12, C18 and C24 were used the same method to extract and merge data, and the spin-up time
of them was the first 12 h, 18 h and 24 h of data, respectively.

### 3.2 ICON test for using the restart files

The meteorological data of the period 12~36 h was cut to estimate the $PM_{2.5}$ concentrations by restart mechanism of CAMx
model. For using the restart files, icbcprep module also used clean initial concentration fields at the beginning of the first-day
simulation. And gridded three-dimensional instantaneous concentrations of all species on all grids were written at the end of
the simulation to allow for a model restart. Then ICON used the 24-h forecast results from the day before as the initial
conditions for the following days shown in Fig. 7 (b). The first day of the simulation starts at 12:00 UTC, and the following
days starts at 00:00 UTC. In order to explore how long is the spin-up time can eliminate the error caused by the initial value,
the sensitivity experiments R1120 and R1124 were set according to the time of the first day for the model simulation, which
began on November 20[th], 2016 and November 24[th], 2016, respectively.

### 3.3 ICON test for continuous simulation

For the continuous simulation, sensitivity experiments CT12 and CT24 were set according to the start time of the intercepted
time periods, which were started at 00:00UTC and 12:00UTC, respectively, shown in Fig. 7 (c). For CT12, the meteorological
data of the period 12~36 h was cut and merged to one file. The period 24~48 h was cut and merged for CT24. Also we built





the continuous emission files by SMOKE model. During the simulation, there was no interruption and finally generated a long-term sequence simulation result for each start time.

## 3.4 Emission test for different emission inventories

Based on the initial condition tests, we selected the best method to do the emission sensitivity experiments. For the emission tests, we compared the simulation results of the original emission inventory (sensitivity experiments Enc) and the updated local emission inventory with the construction fugitive dust emissions (sensitivity experiments Ec).

**Table 3.** The simulation experiment configurations. C00-C24, R1120, R1124, CT12 and CT24 were used to investigate the impact of simulation methods, start time and extracted time period. The impact of different emission inventory was investigated by Ec and Enc. Method C, R, CT presented for the methods of the clean initial condition mechanism, restart mechanism and continuous simulation. Emission inventory nc and c presented for the original emission inventory and the updated local emission inventory with the construction fugitive dust emissions, respectively.

| Experiment | Method | Emission inventory | Start time and extracted time period |
| --- | --- | --- | --- |
| C00 | C | c | 2016/11/26 0-24th hour |
| C06 | C | c | 2016/11/26 6-30th hour |
| C12 | C | c | 2016/11/26 12-36th hour |
| C18 | C | c | 2016/11/26 18-42th hour |
| C24 | C | c | 2016/11/26 24-48th hour |
| R1120 | R | c | 2016/11/20 12-36th hour |
| R1124 | R | c | 2016/11/24 12-36th hour |
| CT12 | CT | c | 2016/11/26 12-36th hour |
| CT24/Ec | CT | c | 2016/11/26 24-48th hour |
| CT24/Enc | CT | nc | 2016/11/26 24-48th hour |





**Figure 7.** The time period for each initial condition experiments. **(a)** shows the time period for C experiments. The output files of CAMx were initialized at 13:00 UTC every day and the CAMx model forecasted the next 48 hours' PM$_{2.5}$ concentrations. The grids with number represent the valid period of each output file. The shaded grids represent the data for one single day, which is extracted by cutting and merging the data of the valid period. **(b)** and **(c)** show the time period for R and CT experiments, respectively.



## 4 Results and discussion

In this study, we collected the observations in December 2016, and evaluate the model performance and improvement. The model ability from both the meteorological field and the daily PM$_{2.5}$ simulations in Xi'an is evaluated in this study.

### 4.1 Model performance of the initial condition tests

There are 13 NSAQ Stations in Xi'an which marked as squares in Fig 2. Nine stations are in urban Xi'an,including GYC, XQ, FZC, XZ, TYC, GXXQ, JKQ, QJ and GYT. Three stations are located in suburban towns, including CAQ, YLQ and LTQ. The CT Station is the city background station, which is in northern urban Xi'an.

### 4.1.1 Sensitivity experiments for using clean initial condition files

Taylor nomogram (Taylor, 2001; Gates et al., 1999) is used to evaluate the accuracy of simulated PM$_{2.5}$ daily concentrations for NSAQ stations which was for the sensitivity experiments of using the clean initial condition files, shown in Fig 8. There are three statistical parameters to evaluate model accuracy, which are the correlation coefficient (R), normalized standard deviation (NSD), normalized root, and mean square error (NRMSE) in Taylor nomogram (Taylor, 2001; Gates et al., 1999; Chang et al., 2004). The sensitivity experiments C00, C06, C12, C18, and C24 were shown by symbols of different colors. We randomly selected 3 stations in urban Xi'an, 2 stations in county towns and a background station to show the simulation results. And the "AVG" meant the average of 13 NSAQ Stations.

As shown in Fig 8, R is 0.36-0.76 for the sensitivity experiments C00, C06, C12, C18, and C24. C24's R is largest and best over all the NSAQ Stations, and C00's is lowest. The NRMSE, which measures the distance from the marker to the REF in Taylor nomogram, is smallest and best for C24, and longest for C00. For NSD, most NSAQ Stations have similar regularity, that is, the NSD values from C00 to C24 are getting closer to "1". The other statistical parameters are presented in Table 4. From experiments C00 to C24, the absolute mean bias (MB) and the mean error (ME) decrease from 51.07 μg/m$^3$ to 3.72 μg/m$^3$ and from 74.09 μg/m$^3$ to 45.82 μg/m$^3$, respectively. The absolute normal mean bias (NMB) and the normal mean error (NME) decrease from 29.73% to 2.17% and from 43.12% to 26.67%, respectively. And the index of agreement (IOA) increase from 0.50 to 0.8. In general, the model performance of C24 is better than other sensitivity experiments in clean initial mechanism tests.

The NSAQ Stations, which in urban city and county towns, have different model performances presented in Fig 8. The model performances of XZ, TYC, XQ in urban city are of poor qualities. On the contrary, YLQ and LTQ in county towns have better model performances than stations in urban city.



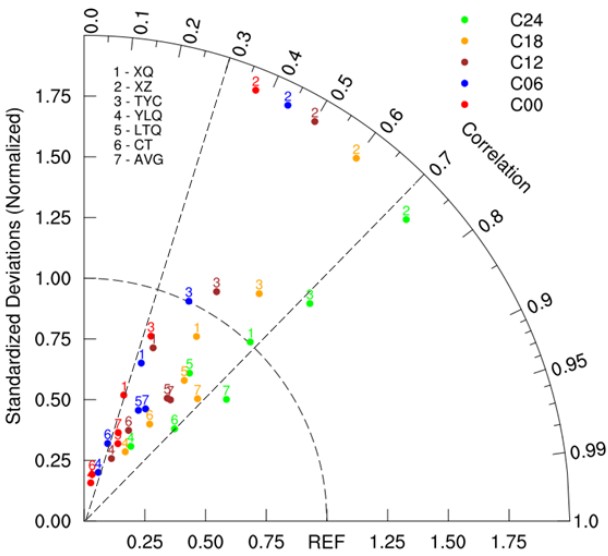

**Figure 8.** Taylor nomogram for modelled and observed daily averaged PM$_{2.5}$ concentrations for the sensitivity experiment of using the clean initial condition files. The "AVG" meant the average of 13 NSAQ Stations. The sensitivity experiments C00, C06, C12, C18, and C24 were shown by symbols of different colors. REF represents a perfect simulated result according to Chang et al. (2004) for air quality model.

### 4.1.2 Sensitivity experiments for using restart files

In order to explore the restart mechanism, sensitivity experiments R1120 and R1124 were set according to the time of the first day for the model simulation. Starting from 12:00 UTC on November 24$^{th}$, the PM$_{2.5}$ concentration simulation results of the two sensitive experiments, R1120 and R1124, were shown in Fig 9. At first, the results of the two sensitivity experiments were very different, and then the two lines were gradually fitted until 16:00 UTC on November 25$^{th}$. After 16:00 UTC on November 25$^{th}$, the two lines fitted almost completely. Therefore, the spin-up time of 27 hours can eliminate the error brought by the initial field for the PM$_{2.5}$ concentrations in CAMx model.

As shown in Table 4, the model performances of the sensitivity experiments R1120 and R1124 are similar in December, 2016. For the sensitivity experiments R1120 and R1124, the R value between observations and simulations is 0.70. The mean bias (MB) and the mean error (ME) are 4.01 μg/m$^3$ and 49.68 μg/m$^3$, respectively. The normal mean bias (NMB) and the normal mean error (NME) are 2.33% and 28.92%, respectively. The value of root mean square error (RMSE) is 67.28 and the IOA reaches 0.82.



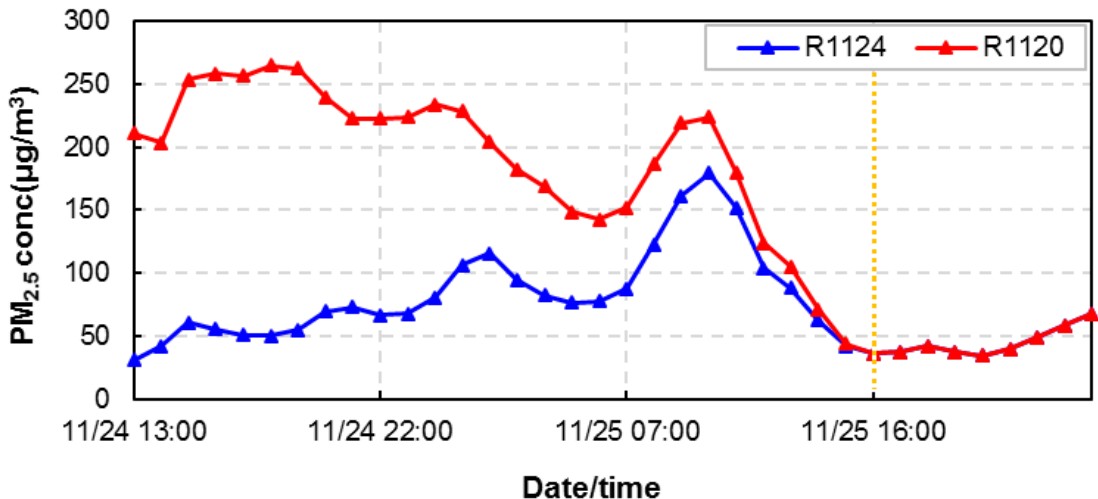

**Figure 9.** The time series of hourly simulated $PM_{2.5}$ concentrations for using the restart files during a period of the spin-up time. The red and blue lines represent the model sensitivity experiments R1120 and R1124, respectively. The begin day of R1120 for model simulation was November 20[th], 2016, and R1124 was November 24[th], 2016.

### 4.1.3 Sensitivity experiments for continuous simulation

As for the continuous simulation, sensitivity experiments CT12 and CT24 were conducted. Although the sensitivity experiments CT12 and R1120 use different methods to generate the initial concentration fields, the start time of the intercepted time periods for the two experiments are the same. $PM_{2.5}$ concentrations of CT12 and R1120 are presented in Fig 10. As shown in Fig 10, the points lie very close to the perfect line "y=x", which indicates that the simulation results of CT12 and R1120 were nearly identical.

The model starting time of sensitivity experiments CT12 and CT24 are November 26[th] 00:00UTC and November 26[th] 12:00UTC, respectively. The concentration accumulation of CT24 is 12 hours more than that of CT12. As shown in Fig 11 there is an air pollution peak in December 2016, which CT24 matches better than CT12. The statistical parameters of CT12 and CT24 are presented in Table 4. The mean bias (MB) and mean error (ME) of CT24 results are 6.29 $\mu g/m^3$ and 42.67 $\mu g/m^3$, respectively, a little better than the CT12 results. The root mean square error (RMSE) of CT24 results is 68.21, also slightly better than the CT12 results. From CT12 to CT24, the R and IOA increase from 0.69 to 0.81 and from 0.81 to 0.90, respectively. Thus, the sensitivity experiments CT24 has better model performance than CT12.



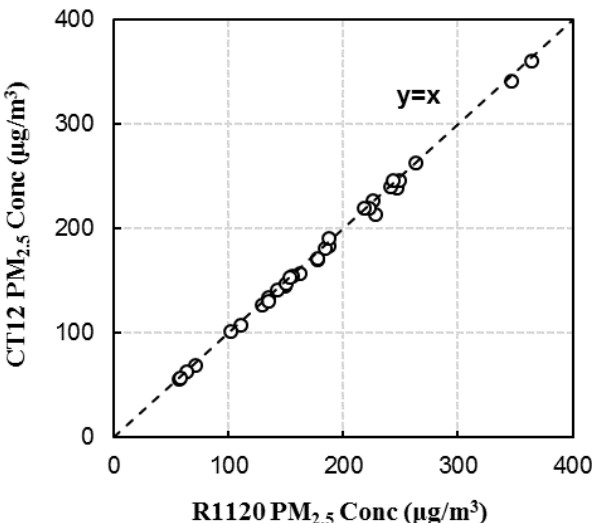

**Figure 10.** Scatter diagram of the R1120 and CT12 experiments of PM₂.₅ concentrations. Line "y=x" represents the simulated
of R1120 is the same to CT12.

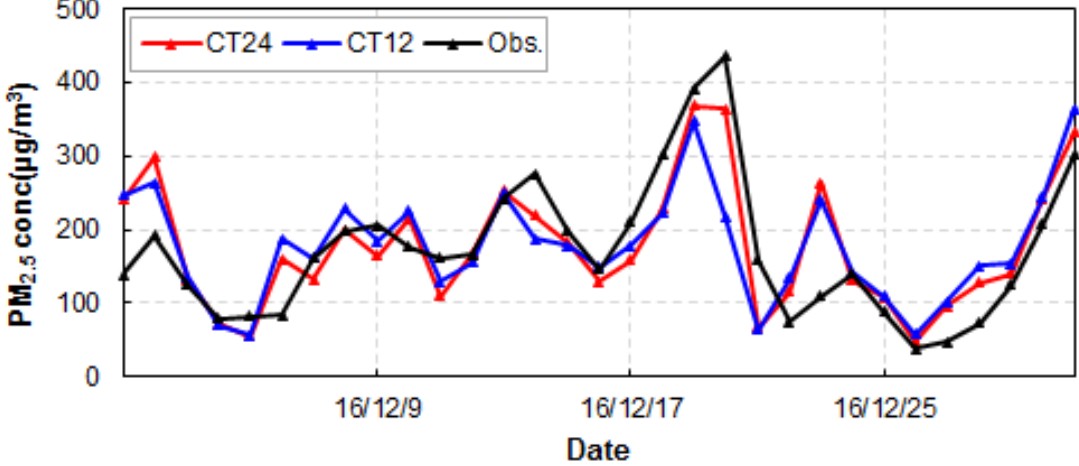

**Figure 11.** The time series of daily PM₂.₅ concentrations for continuous simulation in Xi'an. The black line represents
observations, the blue and red lines show simulated data started at November 26[th] 00:00UTC and November 26[th] 12:00UTC,
respectively.

**4.2 Model performance of emission tests**

Taylor nomogram for modelled and observed daily averaged PM₂.₅ concentrations for all initial condition sensitivity
experiments, shown in Fig 13. The red symbols indicate the sensitivity experiment for using the clean initial condition files,
the blue symbols represent the sensitivity experiment for using the restart files, and the brown symbols show the sensitivity



experiment for continuous simulation. One experiment per symbol. The circles and triangles represent "bias". As shown in Fig
13, R is 0.36 ~ 0.81 in all initial condition sensitivity experiments. R of CT24 is the largest and best in all initial condition
sensitivity experiments. The marker of CT24 has the shortest distance to the "REF" than other initial condition sensitivity
experiments, which means that the NRMSE is the smallest. The NSD of CT24 is 0.92, which determines the modeled and
observed patterns have more consistent amplitude of variation. According to these statistical parameters, the sensitivity
experiments CT24 have the best model performance than other initial condition sensitivity experiments.

Based on the initial condition tests, we selected the best method, CT24, to do the emission sensitivity experiments, shown
in Fig 12. CT24 is the experiment with construction fugitive dust emissions (sensitivity experiments Ec) and the sensitivity
experiments Enc is not. As shown in fig 12, the simulated $PM_{2.5}$ concentrations of Ec play the better model performance than
that of Enc in the high concentration range. As the results shows in Fig 13, R of Ec and Enc are 0.81 and 0.85, respectively.
The NRMSE for Enc is smaller than Ec shown in the taylor nomogram. However, the NSD of Ec, 0.92, is better than Enc,
0.74. And the bias of Enc is much larger than Ec. The other statistical parameters are presented in Table 4. The ME decreased
from 49.18 $\mu g/m^3$ to 42.67 $\mu g/m^3$ and the IOA of simulation results with the updated local emissions reached 0.90. Thus,
compared to the simulation results based on the original emission inventory, a new simulation results, which were driven by
the updated local emissions, showed a much better performance on $PM_{2.5}$ concentrations.

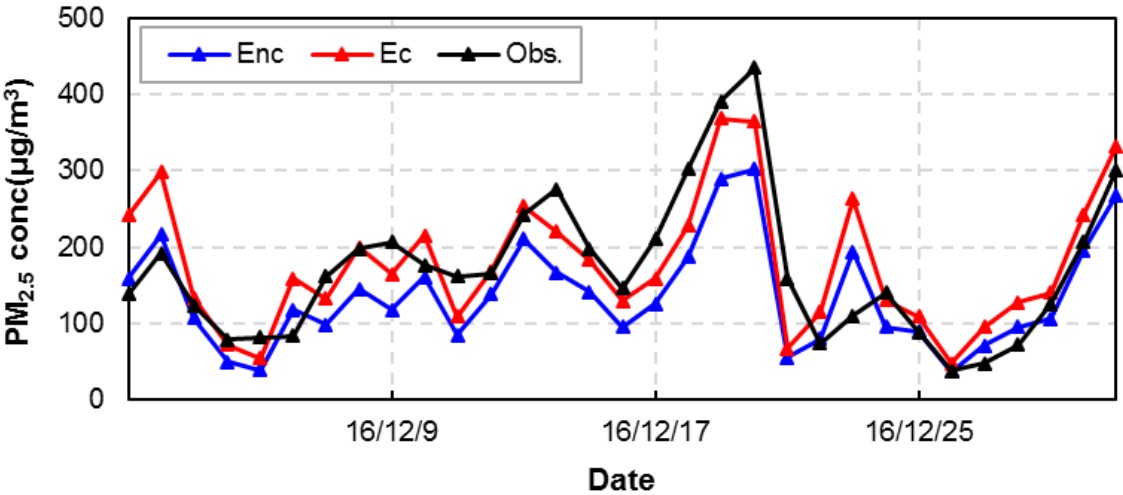

**Figure 12.** The time series of daily observed and simulated $PM_{2.5}$ concentrations averaged from 13 NSAQ Observation Stations
during December 2016 in Xi'an. The black line represents the observations, the blue line represents the simulated by the CAMx
model with construction fugitive dust, and the red line represents the simulated values without construction fugitive dust.



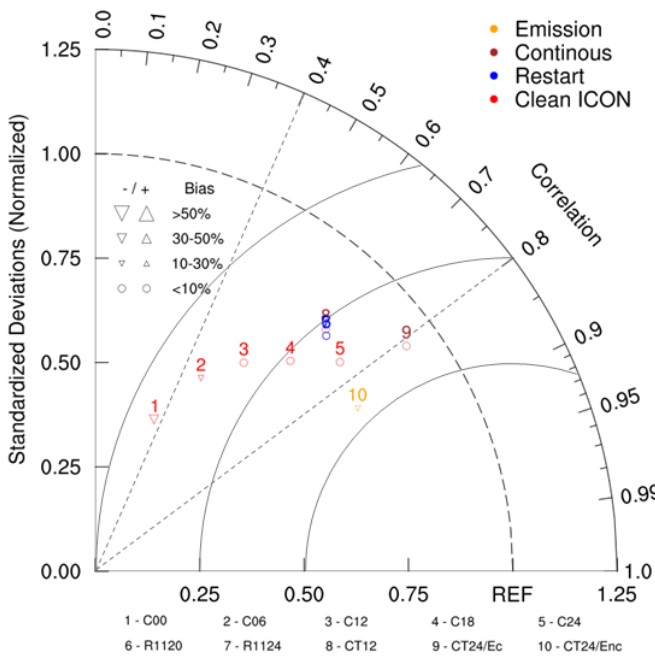

**Figure 13.** Taylor nomogram for modelled and observed daily PM$_{2.5}$ concentrations for all sensitivity experiments under different initial conditions and emissions. The red symbols indicate the clean initial mechanism, the blue symbols represent the restart mechanism, the brown symbols show the sensitivity experiment for continuous simulation, and the orange symbols are for emission tests. The triangles and circles signify "Bias". The scale of the triangle's size represents bias value and the direction of the triangle's vertex represents positive or negative.

**Table 4.** Statistical measures of the modelled daily PM$_{2.5}$ in Xi'an, unit: μg/m.3.

|  | R | MB(μg/m$^3$) | ME(μg/m$^3$) | NMB% | NME% | RMSE | IOA |
|---|---|---|---|---|---|---|---|
| C00 | 0.36 | -51.07 | 74.09 | -29.73 | 43.12 | 100.72 | 0.49 |
| C06 | 0.48 | -24.17 | 60.95 | -14.07 | 35.48 | 85.50 | 0.61 |
| C12 | 0.58 | -12.88 | 53.25 | -7.50 | 30.99 | 76.64 | 0.70 |
| C18 | 0.68 | -7.00 | 48.83 | -4.08 | 28.42 | 68.85 | 0.78 |
| C24 | 0.76 | -3.72 | 45.82 | -2.17 | 26.67 | 60.12 | 0.86 |
| R1120 | 0.70 | 4.01 | 49.68 | 2.33 | 28.92 | 67.28 | 0.82 |
| R1124 | 0.70 | 4.01 | 49.68 | 2.33 | 28.92 | 67.28 | 0.82 |
| CT12 | 0.69 | 6.73 | 50.20 | 3.92 | 29.22 | 68.21 | 0.81 |
| CT24/Ec | 0.81 | 6.29 | 42.67 | 3.66 | 24.83 | 55.29 | 0.90 |
| CT24/Enc | 0.85 | -35.16 | 49.18 | -20.47 | 28.63 | 61.22 | 0.86 |





## 4 Conclusions

The WRF-SMOKE-CAMx model system has been used to simulate the fine particular (PM2.5) concentrations in Xi'an in December, 2016. In this study, the emissions of construction fugitive dust in Xi'an were added in the SMOKE to update the
local emission inventory. A series of model sensitivity experiments for the initial conditions and emissions are designed to improve the model performance in the megacity, Xi'an.

Three methods were applied for the initial condition tests: using the clean initial condition files as clean initial mechanism, using the restart files as restart mechanism and continuous simulation. All initial condition sensitivity experiments are driven by the updated emission inventories. The emission tests are based on the initial condition sensitivity experiment which has the
best model performance.

Comparing the model performance of PM2.5 concentrations in different model sensitivity experiments in Xi'an, we found that the model combining the method of continuous simulation with the updated local emission inventory can nicely improve the model performance. According to statistical parameters, for initial condition tests, the model performance of CT24, C24 and R1120/R1124 are the best. R ranges from 0.36 to 0.81 in all initial condition sensitivity experiments. R of CT24 is the
largest and best in all initial condition sensitivity experiments. R of C24 and R1120/R1124 can reach 0.76, 0.70, respectively. The MB of CT24, C24 and R1120/R1124 are lower, which are 6.29 $\mu g/m^3$, -3.72 $\mu g/m^3$ and 4.01 $\mu g/m^3$, respectively. The IOA of CT24, C24 and R1120/R1124 all reach above 0.8, of which CT24 is 0.9. Compared with other methods, the method of using the clean initial condition files has longer simulation time and larger data volume. Therefore, the method of continuous simulation for hindcast, which is to retrieve PM2.5 concentrations is suggested. For air quality forecast, we can give priority to
the method of restart mechanism. Also, for simulating PM2.5 concentrations by CAMx model, the simulate needs the spin-up time at least 27 hours. This can improve the simulation effect and reduce the simulation time.

This study updated the emissions inventory, which added construction fugitive dust emissions to the original emissions inventory. Compared to the simulation results based on the original emission inventory, a new simulation results, which were driven by the updated local emissions, showed a much better performance on PM2.5 modelling. The absolute MB decreased
from 35.16 $\mu g/m^3$ to 6.29 $\mu g/m^3$ and the IOA of simulation results with the updated local emissions reached 0.90. Therefore, the right addition of emissions will also help to improve the effects of simulation and forecasting.

In finally, we recommend the method of continuous simulation for hindcast, which has the best model performance of PM2.5 concentrations, and can also reduce the output of IO files to improve computing efficiency. For forecast, the method of restart mechanism is suggested, which can reach similar model performance as the continuous simulation. If the restart mechanism
cannot be used due to the limitation of computing resources and storage space when forecasting PM2.5 concentrations, try to extend the spin-up time as much as possible, at least 27 hours according to this work.



**Code and data availability**

The source codes of the WRF model version 3.9.1.1 used in this study are available online at https://www2.mmm.ucar.edu/wrf/users/download/get_source.html(NCAR, 2020, last access: 4 June 2020). The CAMx
version 6.1 code is available at http://www.camx.com/download/default.aspx(ENVIRON, 2020, last access: 4 June 2020), and the SMOKE version 2.4 code is available at https://www.cmascenter.org/smoke/ (CMAS, 2020, last access: 4 June 2020). The global final analysis data (FNL) are from https://rda.ucar.edu/datasets/ds083.2/(NCEP, 2000, last access: 4 June 2020). The dataset related to this manuscript is available online via ZENODO (https://doi.org/10.5281/zenodo.3824676)(Xiao et al., 2020).

**Author contribution**

Han Xiao did the simulation and prepared the materials. Qizhong Wu designed the WRF-SMOKE-CAMx modelling system for Xi'an, including emission processes. Xiaochun Yang collected the local emission inventory in Shaanxi province and help emission processes. Lanning Wang and Huaqiong Cheng help to prepare the model dataset and figure.

**Acknowledgements**

The National Key R&D Program of China (2017YFC0209805), the National Natural Science Foundation of China
(41305121) and the Beijing Advanced Innovation Program for Land Surface funded this work.

**Appendix A**

Statistical parameters for the model evaluation:

Mean bias (MB):

$$MB = \frac{\sum(M_i - O_i)}{n} \qquad (A1)$$

Mean error (ME):

$$ME = \frac{\sum |M_i - O_i|}{n} \qquad (A2)$$

Normalized mean bias (NMB):

$$NMB = \frac{\sum(M_i - O_i)}{\sum O_i} \qquad (A3)$$

Normalized mean error (NME):

$$NME = \frac{\sum |M_i - O_i|}{\sum O_i} \qquad (A4)$$

Root Mean Square Error (RMSE):



$$RMSE = \left[\frac{1}{n}\sum_{i=1}^{n}(M_i - O_i)^2\right]^{\frac{1}{2}} \qquad (A5)$$

Correlation coefficient (R):

$$R = \frac{\sum_{i=1}^{n}(M_i - \bar{M})(O_i - \bar{O})}{\sqrt{\sum_{i=1}^{n}(M_i - \bar{M})^2 \sum_{i=1}^{N}(O_i - \bar{O})^2}} \qquad (A6)$$

Index of agreement (IOA):

$$IOA = 1 - \frac{\sum_{i=1}^{n}(M_i - O_i)^2}{\sum_{i=1}^{n}(|M_i - \bar{O}| + |O_i - \bar{O}|)^2} \qquad (A7)$$

Normalized Standard Deviations (NSD):

$$NSD = \frac{\sqrt{\frac{\sum_{i=1}^{n}(M_i - \bar{M})^2}{n}}}{\sqrt{\frac{\sum_{i=1}^{n}(O_i - \bar{O})^2}{n}}} \qquad (A8)$$

In the equations, $M_i$ and $O_i$ represent the simulated value and observation value of a station, respectively. $n$ represents the

number of stations. $\bar{M}$ and $\bar{O}$ represent the average value of simulated value and observation value, respectively.

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
