# Peer review of "Numerical study of the initial condition and emission on simulating PM2.5 concentrations in Comprehensive Air Quality Model with extensions version 6.1 (CAMx v6.1): Case study of Xi'an"

_Geoscientific Model Development, 2020_

## Referee Comment (RC1) · Anonymous Referee #1 · 5 Jul 2020

In this Manucript, authors employed CAMx to test the impact of initial conditions and fugive dust on PM in Xi'an China. The topic is interesting and helpful to understand the application of chemical transport models. The manuscript is acceptable after addressing the following comments. 1. The initial conditions is very important to model simulation. This manuscript investigated the duration of initial concentrations. The degree of agreement between initial conditions and observations largely the model performance in the first hours. I suggested that the authors compared initial conditions with observations. If possible, the manuscript can design different initial conditions.

2. The impact of initial conditions is different on different species. I suggested that different components (SO4, NO3, NH4, primary PM) or SO2 NOx were discussed.

3. More clear figure captions like figure 7 is needed.

4. Some figures need high PPI like Figure11 and 12.

5. IN figures 12, Why did that simulated PM2.5 with fugive dust emissions.is higher than that with fugive dust emissions.

―――――――――――――――――――

---

## Author Comment (AC1) · 28 Jul 2020

Thank you for your efforts on this manuscript. The comments are valuable and very helpful for revising and improving this work. The detailed responses to the comments are given below one by one.

1. The initial conditions is very important to model simulation. This manuscript investigated the duration of initial concentrations. The degree of agreement between initial conditions and observations largely the model performance in the first hours. I suggested that the authors compared initial conditions with observations. If possible, the manuscript can design different initial conditions.

Response: In our manuscripts, we have compared the simulated and observed PM2.5 concentrations in December 2016. So here we compared the model results in the sensitivity experiments of R1120, R1124, C24 and CT24 with the observation from the time of the simulation started to 00:00 UTC on December 1st, shown in Figure 1. Although the degree of agreement between initial conditions and observations largely the model performance in the first hours, the results of sensitivity experiments for restart mechanism showed that 27 hours of simulation can eliminate the influence of the initial value and improve the simulation results under the same emission and meteorological field. One of the best ways to improve model performance is to consider the assimilation of the initial conditions. But this technology is more difficult, we are still developing.

**Figure 1.** The time series of daily observed and simulated  $PM_{2.5}$  concentrations averaged from 13 NSAQ Observation Stations from the time of the simulation started to 00:00 UTC on December 1st.

**2. The impact of initial conditions is different on different species. I suggested that different components (SO4, NO3, NH4, primary PM) or SO2 NOx were discussed.**

Response: While verifying the model performance of PM2.5 concentrations, this study also verified the model performance of sulfur dioxide (SO2) and nitrogen dioxide (NO2) concentrations, which are the important precursors of SO4 and NO3, parts of particulate matter, based on CAMx simulation under the initial restart mechanism. Figure 2 shows the time series of daily average SO2 and NO2 concentrations and the statistical results are listed in Table 1. The model has an obvious overestimation of SO2, with an average bias of 156.31  $\mu$ g/m3, and the observed SO2 concentration is only 18% of the simulated value. The main reason is that the implementation of desulfurization projects for important emission sources such as coal-fired power plants in recent years has not been fully considered, which has led to overestimation of SO2 emissions in the emission inventory. Li et al. (2017) found that the SO2 emissions

in China have decreased by 75% during the year 2007 to 2016, that is, SO2 emissions in 2016 were about 25% of 2007. If the simulated SO2 concentrations are divided by 4, the statistical parameters will be greatly improved, shown in Table 1. Also the intensity of emissions reduction has uneven spatial distribution. In summary, the overestimation of SO2 is due to the lack of relevant data. The model performance of NO2 concentration is better, the IOA reaches 0.82, and the MB is only  $3.32\mu$ g/m3. There is high consistency of variation trend between the simulated and observed concentrations of SO2 and NO2, with R being 0.81 and 0.75, respectively. The varication of SO2 and NO2 will be added in our revised manuscripts followed this comment.

**Figure 2.** The time series of daily observed and simulated SO2 (top)and NO2 (bottom) concentrations averaged from 13 NSAQ Observation Stations during December 2016 in Xi'an.

| Species         | Mean(µg/m 3 ) |        | R    | MB            | ME            | NMB  | NME  | RMSE   | IOA  |
|-----------------|--------------------------|--------|------|---------------|---------------|------|------|--------|------|
|                 | Obs.                     | Sim.   |      | $(\mu g/m^3)$ | $(\mu g/m^3)$ |      |      |        |      |
| $SO_2$          | 35.45                    | 191.76 | 0.81 | 156.31        | 156.31        | 4.41 | 4.41 | 171.73 | 0.11 |
| $SO_2/4

---

## Referee Comment (RC2) · Anonymous Referee #2 · 31 Aug 2020

Many model sensitivity tests were conducted to assess the effects of different initial conditions and emissions in Xi'an, Northwest China, in December 2016 by using the WRF-CAMx model. Major revisions are needed as suggested below, 1) The simulation was done in the period of November 20, 2016 to January 20, 2017 but the comparison of simulations and observations was made only in December, 2017, why? 2) For model evaluation of meteorological fields, the comparison of simulated and observed wind speed and direction should be added. 3) The CAMx has only one domain . . .(section 2.3)? Actually, Fig.4 shows that the CAMx also has three nested domains. 4) Please

explain why the model performance in the studied urban area is poorer than that in the studied counties (section 4.1.1). 5) Please use "simulation" instead of "mechanism" in the whole text. 6) The writing English need more polishment.

---

## Author Response (AR1)

Thanks a million for your precious time. According to the two referees' comments, we have modified the manuscripts point-by-point, including the followed main aspects:

1) According to Referee 1's comment, we added the model evaluation of $SO_2$ and $NO_2$. Daily observed and simulated $SO_2$ and $NO_2$ concentrations were averaged from 13 National Standard Air Quality Stations during December 2016 in Xi'an. Due to the lack of data on the implementation of desulfurization projects for important emission sources such as coal-fired power plants in recent years, the model has an obvious overestimation of $SO_2$. The model performance of $NO_2$ concentration is better. And there is high consistency of variation trend between the simulated and observed concentrations of $SO_2$ and $NO_2$, with R being 0.81 and 0.75, respectively.

2) We have revised the clearer figure caption for Figure 7.

3) We have increased the PPI of Figure11 and 12.

4) According to Referee 2's comment, we referenced the study of Yang et al. (2020) on wind speed at a 10 m altitude (W10).

5) We deleted the comparison of model performance between some urban stations and suburban stations in the manuscript, and wrote the complete evaluation results of all stations in the supplement.

6) We have used "simulation" instead of "mechanism" in the revised version.

7) Since the reviewer mentioned the language issue, the professional language copy-editing for GMD will be called for the final revised manuscript to solve the language issue. We hope this measure could further improve the presentation quality of the manuscript.

**Response to Referee Comment #1:**

1. The initial conditions is very important to model simulation. This manuscript investigated the duration of initial concentrations. The degree of agreement between initial conditions and observations largely the model performance in the first hours. I suggested that the authors compared initial conditions with observations. If possible, the manuscript can design different initial conditions.

Response: In our manuscripts, we have compared the simulated and observed $PM_{2.5}$ concentrations in December 2016. So here we compared the simulated and observed $PM_{2.5}$ concentrations in the sensitivity experiments of R1120, R1124, C24 and CT24 from the time of the simulation started to 00:00 UTC on December 1$^{st}$, shown in Figure 1. Although the degree of agreement between initial conditions and observations had a large difference in the first hours, the results of sensitivity experiments for restart simulation showed that 27 hours of simulation can eliminate the influence of the initial value and improve the simulation results under the same emission and meteorological field. One of the best ways to improve model performance is to consider the assimilation of the initial conditions. But this technology is more difficult, we are still developing.

[Figure]

**Figure 1.** The time series of daily observed and simulated PM$_{2.5}$ concentrations averaged from 13 NSAQ Observation Stations from the time of the simulation started to 00:00 UTC on December 1$^{st}$.

2. The impact of initial conditions is different on different species. I suggested that different components (SO$_4$, NO$_3$, NH$_4$, primary PM) or SO$_2$ NO$_x$ were discussed.

Response: While verifying the model performance of PM$_{2.5}$ concentrations, this study also verified the model performance of sulfur dioxide (SO$_2$) and nitrogen dioxide (NO$_2$) concentrations, which are the important precursors of SO$_4$ and NO$_3$, parts of particulate matter, based on CAMx simulation under the initial restart mechanism. Figure 2 shows the time series of daily average SO$_2$ and NO$_2$ concentrations and the statistical results are listed in Table 1. The model has an obvious overestimation of SO$_2$, with an average bias of 156.31 µg/m$^3$, and the observed SO$_2$ concentration is only 18% of the simulated value. The main reason is that the implementation of desulfurization projects for important emission sources such as coal-fired power plants in recent years has not been fully considered, which has led to overestimation of SO$_2$ emissions in the emission inventory. Li et al. (2017) found that the SO$_2$ emissions in China have decreased by 75% during the year 2007 to 2016, that is, SO$_2$ emissions in 2016 were about 25% of 2007. If the simulated SO$_2$ concentrations are divided by 4, the statistical parameters will be greatly improved, shown in Table 1. Also the intensity of emissions reduction has uneven spatial distribution. In summary, the overestimation of SO$_2$ is due to the lack of relevant data. The model performance of NO$_2$ concentration is better, the IOA reaches 0.82, and the MB is only 3.32µg/m$^3$. There is high consistency of variation trend between the simulated and observed concentrations of SO$_2$ and NO$_2$, with R being 0.81 and 0.75, respectively. The variation of SO$_2$ and NO$_2$ will be added in our revised manuscripts followed this comment.

[Figure]

**Figure 2.** The time series of daily observed and simulated SO₂ (top)and NO₂ (bottom) concentrations averaged from 13 NSAQ Observation Stations during December 2016 in Xi'an. The black and green lines indicate observed and simulated results, respectively.

**Table 1.** Statistical verification parameters of SO₂ and NO₂ during December 2016in Xi'an.

| Species | Mean(μg/m³) | | R | MB | ME | NMB | NME | RMSE | IOA |
|---------|------|------|------|------|------|------|------|------|------|
| | Obs. | Sim. | | (μg/m³) | (μg/m³) | | | | |
| SO₂ | 35.45 | 191.76 | 0.81 | 156.31 | 156.31 | 4.41 | 4.41 | 171.73 | 0.11 |
| SO₂/4 | | 47.94 | 0.81 | 12.49 | 14.09 | 0.35 | 0.40 | 18.19 | 0.66 |
| NO₂ | 76.77 | 80.09 | 0.75 | 3.32 | 12.86 | 0.04 | 0.17 | 17.13 | 0.82 |

And the corresponding modified contents were added to page 21, line 348 in the revised manuscript, and the reference was added to 26, line 508 as following:

**4.3 Model performance of SO₂ and NO₂**

The sulfur dioxide (SO₂) and nitrogen dioxide (NO₂) concentrations are the important precursors of SO₄ and NO₃, parts of particulate matter. Daily observed and simulated SO₂ and NO₂ concentrations averaged from 13 NSAQ Stations under the initial restart simulation. Figure 14 shows the time series of daily average SO₂ and NO₂ concentrations averaged from 13 NSAQ Stations under the initial restart simulation, and the statistical results are listed in Table 5. The model has an obvious overestimation of SO₂, with an average bias of 156.31 μg/m³, and the observed SO₂ concentration is only 18% of the simulated value. The main reason is that the implementation of desulfurization projects for important emission sources such as coal-fired power plants in recent years has not been fully considered, which has led to overestimation of SO₂ emissions in the emission inventory. Li et al. (2017) found that the SO₂ emissions in China have decreased by 75% during the year 2007 to 2016, that is, SO₂ emissions in 2016 were about 25% of 2007. Also the intensity of emissions reduction has uneven spatial distribution. The model performance of NO₂ concentration is better, the IOA reaches 0.82, and the MB is only 3.32μg/m³.

There is high consistency of variation trend between the simulated and observed concentrations of $SO_2$ and $NO_2$, with R being 0.81 and 0.75, respectively.

[Figure]

**Figure 14.** The time series of daily observed and simulated $SO_2$ (top)and $NO_2$ (bottom) concentrations averaged from 13 NSAQ Stations during December 2016 in Xi'an. The black and green lines indicate observed and simulated results, respectively.

**Table 5.** Statistical verification parameters of $SO_2$ and $NO_2$ during December 2016in Xi'an.

| Species | Mean($\mu g/m^3$) | | R | MB | ME | NMB | NME | RMSE | IOA |
|---|---|---|---|---|---|---|---|---|---|
| | Obs. | Sim. | | ($\mu g/m^3$) | ($\mu g/m^3$) | | | | |
| $SO_2$ | 35.45 | 191.76 | 0.81 | 156.31 | 156.31 | 4.41 | 4.41 | 171.73 | 0.11 |
| $NO_2$ | 76.77 | 80.09 | 0.75 | 3.32 | 12.86 | 0.04 | 0.17 | 17.13 | 0.82 |

Li C, Mclinden C, Fioletov V, et al.: India Is Overtaking China as the World's Largest Emitter of Anthropogenic Sulfur Dioxide, Scientific Reports, 2017, 7(1):14304, https://doi.org/10.1038/s41598-017-14639-8.

3. More clear figure captions like figure 7 is needed.

Response: We have revised the figure caption. And the corresponding modified contents were added to page 12, line 230 in the revised manuscript as following:

[Figure]

**Figure 7.** The time period for each initial condition experiments. **(a)** shows the time period for Clean initial condition (mark C) experiments. The first hour results in the output files of CAMx were at 13:00 UTC every day and the CAMx model forecasted the next 48 hours' $PM_{2.5}$ concentrations in each cycle simulation. The sensitivity experiments C00, C06, C12, C18, and C24 extract different time periods (0~24h, 6~30h, 12~36h, 18~42h and 24~48h, respectively) in each output file as valid data, represented by the grids with number. Each grid represents an hour and the numbers on the grids indicate the hours of the data. The grids with numbers represents the valid time period for each output file. In order to analyze from 0:00 Beijing time (16:00 UTC) every day, the 24-hour data of a day is cut and merged from 16:00 UTC in the valid time period of each output file. And the shaded grids represent the data for one single day. **(b)** shows the time period for Restart (mark R) experiments. The meteorological data of the period 12~36 h was cut to estimate the $PM_{2.5}$ concentrations by restart mechanism. The first day of the simulation starts at 12:00 UTC, and the following days starts at 00:00 UTC. **(c)** show the time period for continuous simulation (mark CT) experiments. The meteorological data of the period 12~36 h is cut and merged to one file for CT12 and the period 24~48 h was cut and merged for CT24.

4. Some figures need high PPI like Figure11 and 12.

Response: We have increased the PPI of Figure 11 and 12. The new figure 11 was added to page 17, line 311, and the new figure 12 was add to page 19, line 335 in the revised manuscript as following:

[Figure]

**Figure 11.** The time series of daily PM$_{2.5}$ concentrations for continuous simulation in Xi'an. The black line represents observations, the blue and red lines show simulated data started at November 26[th] 00:00UTC and November 26[th] 12:00UTC, respectively.

[Figure]

**Figure 12.** The time series of daily observed and simulated PM$_{2.5}$ concentrations averaged from 13 NSAQ Observation Stations during December 2016 in Xi'an. The black line represents the observations, the blue line represents the simulated by the CAMx model with construction fugitive dust, and the red line represents the simulated values without construction fugitive dust.

5. IN figures 12, Why did that simulated PM2.5 with fugitive dust emissions.is higher than that without fugitive dust emissions.

Response: Thank you for your comment. The concentrations of PM$_{2.5}$ in the model is the sum of multiple types of particulate matter, including 4 primary particulate matter (PEC, POA, FPRM and FCRS). The construction fugitive dust emissions contain primary particulate matter. So the concentrations of primary particulate matter with construction fugitive dust emissions is higher than that not including building dust emissions, resulting the total concentration of PM$_{2.5}$ with fugitive dust is higher.

**Response to Referee Comment #2:**

1) The simulation was done in the period of November 20, 2016 to January 20, 2017 but the comparison of simulations and observations was made only in December, 2016, why?

Response: According to the study of Yang et al. (2020), December 2016 was a month with severe PM$_{2.5}$ pollution, including a severe pollution process. Therefore, this study finally selected December 2016 for simulation and analysis. Because we did not know how long the spin-up time would take before the simulation experiments, we simulated forward 10 days as the spin-up time.

2) For model evaluation of meteorological fields, the comparison of simulated and observed wind speed and direction should be added.

Response: The comparison of simulated and observed wind speed has been done in the study of Yang et al. (2020). We used the same model configuration and monitoring sites. The simulation time period in the study of Yang et al. (2020) includes the simulation time period of this study. So the model performance was similar. As shown in Fig. 1 (Yang et al., 2020), the WRF detected the variation of meteorological elements in the pollution period. And the verification statistics were shown in Table 1(Yang et al., 2020). As the results show, the W10 is underestimated. The MB of the W10 is -0.14 m/s. The R of the W10 is 0.63, which indicates a good agreement between the observations and the model results. Owing to the particular topography of Xi'an, the wind speed is low at all times. The RMSE of the simulated W10 is only 0.96.

Due to the lack of wind direction data, and the large error in model performance of wind direction shown from previous studies, this study did not evaluate the wind direction.

[Figure]

**Figure 1**. Time series of daily observed and simulated temperatures at a 2 m altitude (T2), relative humidity at a 2 m altitude (RH2) among nine monitoring sites, and daily average wind speed at a 10 m altitude (W10) in Xi'an station from 20 November 2016 to 20 January 2017. The black and red lines represent the observations and the values simulated by WRF. (For interpretation of the references to colour in this figure legend, the reader is referred to the web version of this article.) (Yang et al., 2020)

**Table 1.** Verification statistics of daily temperature at a 2 m altitude (T2), relative humidity at a 2 m altitude (RH2), and wind speed at 10 m altitude (W10) from 20 November 2016 to 20 January 2017. ME, MB, R, and RMSE are abbreviations for mean error, mean bias, correlation coefficient, root-mean-square error, and accuracy rate, respectively. The units of the Mean, ME, and MB are for T2, % for RH2, and m s$^{-1}$for W10. (Yang et al., 2020)

| Variable | Mean | | ME | MB | R | RMSE |
|---|---|---|---|---|---|---|
| | Obs. | Sim. | | | | |
| T2 (℃) | 3.06 | 0.28 | 2.80 | -2.77 | 0.82 | 3.14 |
| RH2 (%) | 69.80 | 79.05 | 9.91 | 9.26 | 0.71 | 11.87 |
| W10 (m·s-1) | 1.98 | 1.83 | 0.79 | -0.14 | 0.63 | 0.96 |

And the corresponding modified contents were added to the page 4, line 102 in the revised manuscript as following:

Under the same model configuration on WRF model and the same verified sites, Yang et al. (2020) compared the simulated and observed wind speed at a 10 m altitude (W10) in Xi'an station from 20 November 2016 to 20 January 2017. As the results show, the W10 is underestimated. The MB of the W10 is -0.14 m/s. The R of the W10 is 0.63, which indicates a good agreement between the observations and the model results.

3) The CAMx has only one domain . . .(section 2.3)? Actually, Fig.4 shows that the CAMx also has three nested domains.

Response: Fig. 4 shows the model domain in the WRF-CAMx modelling system. In this study, three-nest domains were designed for the WRF model, as shown in Figure 4. If the meteorological field only simulates one domain, directly interpolating from the initial field resolution to a 3 km-grid, the error will be relatively large. The CAMx in this study has only one domain and the settings are the same as those in the D3 domain.

4) Please explain why the model performance in the studied urban area is poorer than that in the studied counties (section 4.1.1).

Response: Thank you for your comment. We randomly selected three monitoring sites in the urban area for analysis shown in Fig 8, so the conclusion may not be rigorous enough. According to your suggestion, we checked again and verified the statistical parameters of all monitoring sites. As shown in Table 2, for correlation coefficients (R), the model performances of stations in suburban counties are better than that of some stations in urban city, indicating that the simulation of suburban counties performs better in the aspect of trends. The reason may be that PM2.5 emissions in urban areas are affected by more complex factors, and it is more difficult to simulate weather in urban areas. For bias, the model performances of stations in urban cities are better than that of the suburbs, and the simulated PM$_{2.5}$ concentrations in the suburbs are obviously underestimated. We also used the fraction of predictions within a factor of two of observations (FAC2) to verified the model performances of all the monitoring sites. The FAC2 of the urban cities is 74%, while the FAC2 of the suburban counties is 62%, indicating that the model performance of the urban area is better than that of the suburban area in terms of total emissions. We will revise the manuscript based on the

following results.

**Table 2**. Verification statistics of PM2.5 concentrations on December 2016 among all monitoring sites.

|  | Station | R | MB ($\mu g/m^3$) | ME ($\mu g/m^3$) | NMB % | NME % | RMSE | FAC2 % |
|---|---|---|---|---|---|---|---|---|
| urban | CT | 0.70 | -58.86 | 66.22 | -38.08 | 42.84 | 89.99 | 64 |
|  | XQ | 0.53 | 61.72 | 81.41 | 36.00 | 47.48 | 108.31 | 75 |
|  | JKQ | 0.69 | -76.00 | 84.91 | -37.55 | 41.95 | 109.89 | 76 |
|  | TYC | 0.66 | 75.46 | 89.90 | 45.98 | 54.78 | 108.49 | 77 |
|  | GYC | 0.73 | 19.42 | 52.51 | 11.17 | 30.21 | 68.01 | 89 |
|  | QJ | 0.59 | 14.79 | 71.73 | 8.06 | 39.11 | 89.90 | 76 |
|  | GYT | 0.62 | -31.46 | 69.01 | -18.72 | 41.06 | 84.15 | 66 |
|  | FZC | 0.67 | 3.11 | 60.68 | 1.79 | 35.01 | 76.42 | 78 |
|  | XZ | 0.57 | 51.62 | 79.66 | 32.64 | 50.36 | 101.86 | 72 |
|  | GX | 0.64 | 67.11 | 88.46 | 39.42 | 51.96 | 104.45 | 69 |
| suburban | CAQ | 0.68 | 8.51 | 54.49 | 5.36 | 34.29 | 72.26 | 86 |
|  | YLQ | 0.60 | -97.63 | 99.12 | -57.67 | 58.55 | 120.78 | 31 |
|  | LTQ | 0.57 | -54.43 | 76.28 | -33.53 | 46.98 | 97.49 | 69 |

We deleted the comparison of model performance between some urban stations and suburban stations in the revised manuscript on page 14, line 270, and the corresponding modified contents were added in the **supplement** as following:

Table 1 shows the statistical results of the daily average $PM_{2.5}$ concentrations of 13 NSAQ Observation Stations during December 2016 in Xi'an. The correlation coefficients (R) of most urban stations can reach above 0.6, of which GYC is 0.73, while the R of XQ, QJ and XZ are low. Among the three stations in suburban counties, CAQ had the highest correlation coefficient of 0.68, and the other two stations are lower than 0.6. For bias, the model performances of stations in urban cities are better than that of the suburbs, and the simulated $PM_{2.5}$ concentrations in the suburbs are obviously underestimated. Except for CT, GYT and XZ, the FAC2 of the other 7 urban stations all reach more than 70%. Among the three suburban stations the CAQ has the best model performance, with FAC2 at 86%, while the other two stations are relatively low, especially the YLQ is only 31%. The average FAC2 of the urban cities is 74%, while the average FAC2 of the suburban counties is 62%, indicating that the model performance of the urban area is better than that of the suburban area in terms of total emissions.

**Table 1**. Verification statistics of PM2.5 concentrations on December 2016 among all monitoring sites.

| Station | R | MB | ME | NMB | NME | RMSE | FAC2 |
|---|---|---|---|---|---|---|---|

|          |     |      | (μg/m³) | (μg/m³) | %      | %     |        | %   |
|----------|-----|------|---------|---------|--------|-------|--------|-----|
| urban    | CT  | 0.70 | -58.86  | 66.22   | -38.08 | 42.84 | 89.99  | 64  |
|          | XQ  | 0.53 | 61.72   | 81.41   | 36.00  | 47.48 | 108.31 | 75  |
|          | JKQ | 0.69 | -76.00  | 84.91   | -37.55 | 41.95 | 109.89 | 76  |
|          | TYC | 0.66 | 75.46   | 89.90   | 45.98  | 54.78 | 108.49 | 77  |
|          | GYC | 0.73 | 19.42   | 52.51   | 11.17  | 30.21 | 68.01  | 89  |
|          | QJ  | 0.59 | 14.79   | 71.73   | 8.06   | 39.11 | 89.90  | 76  |
|          | GYT | 0.62 | -31.46  | 69.01   | -18.72 | 41.06 | 84.15  | 66  |
|          | FZC | 0.67 | 3.11    | 60.68   | 1.79   | 35.01 | 76.42  | 78  |
|          | XZ  | 0.57 | 51.62   | 79.66   | 32.64  | 50.36 | 101.86 | 72  |
|          | GX  | 0.64 | 67.11   | 88.46   | 39.42  | 51.96 | 104.45 | 69  |
| suburban | CAQ | 0.68 | 8.51    | 54.49   | 5.36   | 34.29 | 72.26  | 86  |
|          | YLQ | 0.60 | -97.63  | 99.12   | -57.67 | 58.55 | 120.78 | 31  |
|          | LTQ | 0.57 | -54.43  | 76.28   | -33.53 | 46.98 | 97.49  | 69  |

5) Please use "simulation" instead of "mechanism" in the whole text.

Response: The authors thanks for your constructive suggestion. We will use "simulation" instead of "mechanism" in the revised version.

6) The writing English need more polishment.

Response: We will find the experts to help polish the writing English, and try our best to improve the level of writing English.

[revised manuscript text omitted]
 correlation coefficients (R) of most urban stations can reach above 0.6, of which GYC is 0.73, while the R of XQ, QJ and XZ are low. Among the three stations in suburban counties, CAQ had the highest correlation coefficient of 0.68, and the other two stations are lower than 0.6. For bias, the model performances of stations in urban cities are better than that of the suburbs, and the simulated PM$_{2.5}$ concentrations in the suburbs are obviously underestimated. Except for CT, GYT and XZ, the FAC2 of the other 7 urban stations all reach more than 70%. Among the three suburban stations the CAQ has the best model performance, with FAC2 at 86%, while the other two stations are relatively low, especially the YLQ is only 31%. The average FAC2 of the urban cities is 74%, while the average FAC2 of the suburban counties is 62%, indicating that the model performance of the urban area is better than that of the suburban area in terms of total emissions.

**Table S1**. Verification statistics of PM2.5 concentrations on December 2016 among all monitoring sites.

|          | Station | R    | MB ($\mu g/m^3$) | ME ($\mu g/m^3$) | NMB % | NME % | RMSE  | FAC2 % |
|----------|---------|------|--------|--------|--------|-------|--------|------|
|          | CT      | 0.70 | -58.86 | 66.22  | -38.08 | 42.84 | 89.99  | 64   |
|          | XQ      | 0.53 | 61.72  | 81.41  | 36.00  | 47.48 | 108.31 | 75   |
|          | JKQ     | 0.69 | -76.00 | 84.91  | -37.55 | 41.95 | 109.89 | 76   |
|          | TYC     | 0.66 | 75.46  | 89.90  | 45.98  | 54.78 | 108.49 | 77   |
| urban    | GYC     | 0.73 | 19.42  | 52.51  | 11.17  | 30.21 | 68.01  | 89   |
|          | QJ      | 0.59 | 14.79  | 71.73  | 8.06   | 39.11 | 89.90  | 76   |
|          | GYT     | 0.62 | -31.46 | 69.01  | -18.72 | 41.06 | 84.15  | 66   |
|          | FZC     | 0.67 | 3.11   | 60.68  | 1.79   | 35.01 | 76.42  | 78   |
|          | XZ      | 0.57 | 51.62  | 79.66  | 32.64  | 50.36 | 101.86 | 72   |
|          | GX      | 0.64 | 67.11  | 88.46  | 39.42  | 51.96 | 104.45 | 69   |
|          | CAQ     | 0.68 | 8.51   | 54.49  | 5.36   | 34.29 | 72.26  | 86   |
| suburban | YLQ     | 0.60 | -97.63 | 99.12  | -57.67 | 58.55 | 120.78 | 31   |
|          | LTQ     | 0.57 | -54.43 | 76.28  | -33.53 | 46.98 | 97.49  | 69   |

**Appendix SA**

Statistical parameters for the model evaluation:

Fraction of predictions within a factor of two of observations(FAC2):

$$0.5 \leq \frac{M_i}{O_i} \leq 2.0 \qquad (SA1)$$

---

## Author Response (AR2)

**Response to Topical Editor:**

Comments to the Author: Thank you for the responses to the reviewers. I see you have answered them appropriately except the comment concerning English language. Before accepting the manuscript to GMD the language needs to be improved. There are several points (in old and new text) where the language is not good. I recommend the manuscript to be read by a native speaker or a person with similar skills.

Dear Editor,

Thanks a million for your precious time and your suggestion. In order to improve the English language of the manuscript, we called for the English language services from Elsevier. The certificate and the mark-up manuscript version as followed.

[Figure]

[revised manuscript text omitted]
. (2019) a…dded local datasets tointo…the emission inventory of the Guanzhong Plain (, …hina),…which was applied to simulatein simulating…PM$_{2.5}$ concentrations using theby…CMAQ model in Xi'an. Numerous studies haveworks…indicated that construction dust emissions playemission plays…an important role inof the…air pollution, especially in urban areas (Ni et al., 2012; Huang et al., 2014; Wang et al., 2015). In our previous study, we built a particulate matter emission inventory from construction activities at the county level in ... [3]

[revised manuscript text omitted]


[Figure]

**Figure 4.** Three-nest model domain with 27-9-3 km horizontal resolution in the WRF-CAMx modelling system. D1 covers most parts of China, with 148 × 121 grids, and D2 includes Shaanxi, Shanxi and Henan Provinces. The inner domain covers Fen-wei Plain, including Xi'an.

**2.4 Emission Inventory and Processes**

The SMOKE version 2.4 (Houyoux and Vukovich, 1999) model was used to improve the Fen-wei emissions, especially Xi'an local emissions, and provide gridded emissions for the CAMx model in this study. Based on the emission inventories of a previous study (Yang et al., 2019), this study added the emission quantity of $PM_{2.5}$ from construction fugitive dust in Xi'an to update the local emission inventories. The emission inventories in this study include:

1. The regional emissions in East Asia and the local emissions in the Guanzhong Plain were obtained from Wu et al. (2014) and Yang et al. (2019). Major industrial emissions were slightly adjusted according to the annual report in this study. The emission inventory at the city-level is presented in Table 2.

2. Construction fugitive dust emissions in Xi'an, based on the survey data of construction projects in Fig. 5, were collected in a previous study (Xiao et al., 2019), indicated as a "local area source". This is a new dataset at the county level and updated in 2017. The basic data included the location and area of each construction project. We also replenished the missing construction data and corrected the error information with Google Earth and other geographic information tools to obtain more accurate location information. According to statistics, there were 1595 construction projects in Xi'an in 2017, with 86.1 $km^2$ of the total construction area. The construction area in the main urban area (Xincheng, Beilin, Lianhu, Yanqiao, Weiyang, and Yanta) was about 62.2 $km^2$, accounting for 7.5% of the total area in the main urban area. The distribution of the construction fugitive dust emissions in Xi'an is shown in Fig. 6.

[revised manuscript text omitted]

[revised manuscript text omitted]


[Figure]

**Figure 9.** The time series of hourly simulated PM$_{2.5}$ concentrations using the restart files during a spin-up time period. The red and blue lines represent the model sensitivity experiments R1120 and R1124, respectively. The starting day of the model simulation for R1120 was 20$^{th}$ November 2016, and for R1124 was 24$^{th}$ November 2016.

**4.1.3 Sensitivity experiments for continuous simulation**

For the continuous simulation, sensitivity experiments with CT12 and CT24 were conducted. Although the sensitivity experiments CT12 and R1120 use different methods to generate the initial concentration fields, the start times of the intercepted time periods for the two experiments were the same. The PM$_{2.5}$ concentrations of CT12 and R1120 are presented in Fig 10. As shown in Fig 10, the points lie very close to the perfect line "y=x", which indicates that the simulation results of CT12 and R1120 were nearly identical.

The model starting time of sensitivity experiments CT12 and CT24 are 26$^{th}$ November at 00:00 UTC and 26$^{th}$ November at 12:00 UTC, respectively. The concentration accumulation of CT24 was 12 h higher than that of CT12. As shown in Fig 11, there is an air pollution peak in December 2016, in which CT24 matches better than CT12. The statistical parameters of CT12 and CT24 are presented in Table 4. The mean bias (MB) and mean error (ME) of CT24 results were 6.29 µg/m$^3$ and 42.67 µg/m$^3$, respectively, which are slightly better than the CT12 results. The root mean square error (RMSE) of the CT24 results is 68.21, which is also slightly better than the CT12 results. From CT12 to CT24, the R and IOA increased from 0.69 to 0.81 and from 0.81 to 0.90, respectively. Thus, the sensitivity experiments with CT24 have better model performance than CT12.


[revised manuscript text omitted]


from 35.16 µg/m³ to 6.29 µg/m³ and the IOA of simulation results with the updated local emissions was 0.90. Therefore, the right addition of emissions will also help to improve the effects of simulation and forecasting.

Finally, we recommend the continuous simulation method for hindcast, which performs best for PM$_{2.5}$ concentrations, and can also reduce the output of IO files to improve computing efficiency. For forecasting, the method of restart simulation is suggested, which can reach a similar model performance as the continuous simulation. If the restart simulation cannot be used owing to the limitation of computing resources and storage space when forecasting PM$_{2.5}$ concentrations, we attempt to extend the spin-up time as much as possible, at least 27 h according to our results.

**Code and data availability**

The source codes of the WRF model version 3.9.1.1 used in this study are available online at https://www2.mmm.ucar.edu/wrf/users/download/get_source.html(NCAR, 2020, last access: 4 June 2020). The CAMx version 6.1 code is available at http://www.camx.com/download/default.aspx(ENVIRON, 2020, last access: 4 June 2020), and the SMOKE version 2.4 code is available at https://www.cmascenter.org/smoke/ (CMAS, 2020, last access: 4 June 2020). The global final analysis data (FNL) were obtained from https://rda.ucar.edu/datasets/ds083.2/(NCEP, 2000, last access: 4 June 2020). The dataset related to this manuscript is available online via ZENODO (https://doi.org/10.5281/zenodo.3824676)(Xiao et al., 2020).

**Author contribution**

Han Xiao conducted the simulation and prepared the materials. Qizhong Wu designed the WRF-SMOKE-CAMx modelling system for Xi'an, including emission processes. Xiaochun Yang collected the local emission inventory in Shaanxi Province, and assisted in the emission processes. Lanning Wang and Huaqiong Cheng helped prepare the model dataset and figure.

**Acknowledgements**

The National Key R&D Program of China (2017YFC0209805), the National Natural Science Foundation of China (41305121), and the Beijing Advanced Innovation Program for Land Surface funded this work.

**Appendix A**

Statistical parameters for the model evaluation:

Mean bias (MB):

$$MB = \frac{\sum(M_i - O_i)}{n} \qquad (A1)$$



[revised manuscript text omitted]